

# Antigenic and mutational insights into the Nipah virus G glycoprotein: implications for viral entry, host specificity, therapeutics, and vaccine development

Nur Syafiqah Mohamad Nasir[1], Yasmin Khairani Muhammad Ismadi[2], Noreafifah Semail[1], Wan Alif Syazwani Wan Alias[1], Nik Mohd Noor Nik Zuraina[1,3], Nik Yusnoraini Yusof[4], Zakuan Zainy Deris[1,3] and Mohd Zulkifli Salleh[1]

[1] Department of Medical Microbiology & Parasitology, School of Medical Sciences, Universiti Sains Malaysia, Kubang Kerian, Kelantan, Malaysia
[2] Bacteriology Unit, Infectious Disease Research Centre, Institute for Medical Research (IMR), National Institutes of Health, Shah Alam, Selangor, Malaysia
[3] Hospital Pakar Universiti Sains Malaysia, Kubang Kerian, Kelantan, Malaysia
[4] Institute for Research in Molecular Medicine (INFORMM), Health Campus, Universiti Sains Malaysia, Kubang Kerian, Kelantan, Malaysia

## ABSTRACT

Nipah virus (NiV), a highly lethal RNA virus from the *Paramyxoviridae* family, causes severe neurological and respiratory diseases in humans. First identified during the 1990s outbreak in Malaysia, NiV remains a significant global health threat due to the absence of approved vaccines or antiviral treatments. Since its discovery, more than 754 cases have been reported, with a mortality rate exceeding 50%. Despite its classification as a biosafety level 4 pathogen, the molecular mechanisms underlying NiV pathogenesis remain poorly understood. Two surface glycoproteins—the attachment (G) and fusion (F) proteins—play crucial roles in facilitating early stages of cell entry and determining host specificity. While naturally occurring mutations in the G glycoprotein are limited, experimental studies involving engineered mutations have provided critical insights into receptor binding, fusion activation, and immune evasion. This review summarizes current knowledge of these antigenic and mutational findings, highlighting their implications for viral entry and host specificity, and providing valuable insights for the development of vaccines and therapeutics.

# INTRODUCTION

Nipah virus (NiV) is a highly pathogenic re-emerging member of the *Henipavirus* genus in the *Paramyxoviridae* family. The virus was first discovered in 1998 during an outbreak in Sungai Nipah, Malaysia, affecting pig farmers. The outbreak was linked to the transmission

Corresponding authors
Zakuan Zainy Deris, zakuan@usm.my
Mohd Zulkifli Salleh, m.z.salleh@usm.my

of the virus from infected pigs to humans, highlighting its zoonotic nature (*Van Doremalen et al., 2022*). Natural reservoir of NiV includes fruit bats, specifically pteropid bats (flying foxes) under the genus of *Pteropus* (*Chua et al., 2002*; *Mougari et al., 2022*). The virus can be transmitted through contaminated food exposed to bat body fluids, as well as through direct person-to-person contact (*Chadha et al., 2006*; *Gurley et al., 2007*). Infected individuals may experience asymptomatic infection, acute respiratory illness, or, in severe cases, fatal encephalitis (*WHO, 2018*). Globally, over 754 cases of NiV infection have been reported, with mortality rates exceeding 50%, primarily in Bangladesh, India, Malaysia, the Philippines, and Singapore (*Khan et al., 2024*). NiV outbreaks have persisted, occurring annually, particularly in Bangladesh and eastern India, with infections often linked to the consumption of raw date palm sap contaminated with bat saliva or urine. Moreover, these regions report high fatality rates, with Bangladesh accounting for the highest number of cases and a 56% mortality as of May 2024 (*WHO, 2018*; *Khan et al., 2024*). While NiV has shown limited sustained human-to-human transmission compared to respiratory pathogens such as influenza or coronaviruses, its potential for adaptation and evolution remains a significant concern. Despite ongoing outbreaks and high mortality rates, the molecular mechanisms of NiV pathogenesis remain poorly understood. NiV, like other RNA viruses, could have a high mutation rate, facilitating its adaptation to diverse hosts and environments, and contributing to its broad species tropism (*Devnath & Masud, 2021*; *Quarleri, Galvan & Delpino, 2022*; *Skowron et al., 2022*). However, detailed information on the mutation frequency of the NiV G glycoprotein in natural populations remains limited. This review provides an in-depth analysis of structural data on engineered mutations in the NiV G glycoprotein, offering crucial insights into how these changes affect its binding to ephrin-B2 and ephrin-B3 host cell receptors. This review highlights how engineered mutations and conformation-specific antibodies modulate structural dynamics of the G glycoprotein, revealing critical determinants of viral entry and host specificity, with experimental models offering key insights for vaccine and therapeutic development despite the low prevalence of natural variants.

## Rationale for the study

The emergence and persistence of NiV as a high-fatality zoonotic pathogen underscore an urgent need to understand its molecular mechanisms of entry and host adaptation. With no licensed vaccines or antivirals currently available, and the virus classified as a biosafety level 4 (BSL-4) pathogen, preventive strategies hinge on deciphering the viral components that facilitate infection. The surface glycoprotein G plays a pivotal role in receptor recognition and host specificity, making it a prime target for therapeutic intervention. However, natural variation and engineered modifications of the G glycoprotein remain undercharacterized. Moreover, despite extensive characterization of the NiV G head and stalk domains (*Bowden et al., 2008*), structural information on the $\beta$-neck domain, the linker region, and the transmembrane helix remains limited. This gap likely results from the technical challenges of resolving membrane-anchored regions in glycoproteins (*Lee, Fusco & Saphire, 2009*) and limited accessibility to BSL-4 facilities required for functional studies (*Mire et al., 2016*). These underexplored domains offer promising avenues for future investigations into viral

assembly, stability, and host interaction. This review addresses this critical knowledge gap by compiling and analyzing engineered mutations in the NiV G glycoprotein, drawing connections to their implications in viral entry and host adaptation, which would benefit researchers for the development of vaccines and therapeutics.

## SURVEY METHODOLOGY

Relevant literature was identified through searches in PubMed and Google Scholar, using the following keywords: Henipavirus G glycoprotein; Nipah virus G glycoprotein; mutations; mutagenesis; viral entry; host specificity; NiV vaccines; NiV therapeutics. No strict timeframe was applied, but emphasis was placed on recent advancements. To gain structural and functional insights, additional searches included the terms: X-ray crystallography; cryo-electron microscopy; computational modeling studies; animal modeling studies; neutralizing antibodies; subunit vaccines; monoclonal antibodies therapy. Advance searches were further refined using Boolean operators adapted to each platform's requirements. The following Boolean string was used to enhance retrieval of relevant studies, ("Henipavirus G glycoprotein" OR "Nipah virus G glycoprotein") AND (mutation* OR mutagenesis) AND ("viral entry" OR "host specificity") AND ("NiV vaccine*" OR "NiV therapeutic*") AND ("X-ray crystallography" OR "cryo-electron microscopy" OR "computational modeling" OR "animal model*" OR "neutralizing antibody*" OR "subunit vaccine*" OR "monoclonal antibody therapy*") (accessed on 21 January 2025). In addition, the inclusion criteria were limited to English-language articles that specifically investigated mutations in the G glycoprotein of Henipaviruses, particularly NiV. Studies that did not meet these criteria were excluded. These searches aimed to assess how modifications in the NiV G affect receptor binding and viral entry, ultimately evaluating NiV G-based vaccine strategies and therapeutic interventions. Additional sources, including preprints (bioRxiv, medRxiv), and reports from organizations such as the World Health Organization (WHO) and the Centers for Disease Control and Prevention (CDC), were reviewed to incorporate current perspectives. Findings are synthesized to provide a comprehensive understanding of NiV G's structural and functional significance, highlighting key knowledge gaps and future research directions in vaccine and therapeutic development.

### NiV structures

NiV is an enveloped, non-segmented virus with an 18.2-kb negative-sense single-stranded RNA genome that encodes six structural proteins, the surface glycoprotein (G), fusion (F) protein, nucleocapsid (N), matrix protein (M), phosphoprotein (P), and viral polymerase (L) (Fig. 1). Additionally, it encodes three non-structural proteins, C, V, and W, which are expressed in infected cells and produced from the P gene (*Aditi & Shariff, 2019*; *Lawrence & Escudero-Pérez, 2022*). The two NiV surface glycoproteins, attachment G and fusion F, work in tandem to facilitate the early stages of cell entry. The tetrameric G glycoprotein specifically binds to the cell surface receptors ephrin-B2 or ephrin-B3 (*Xu et al., 2008*; *Bender et al., 2016*), triggering conformational changes in the trimeric fusion F glycoprotein and facilitating the fusion of the viral and host cell membranes (Fig. 1) (*Chang*

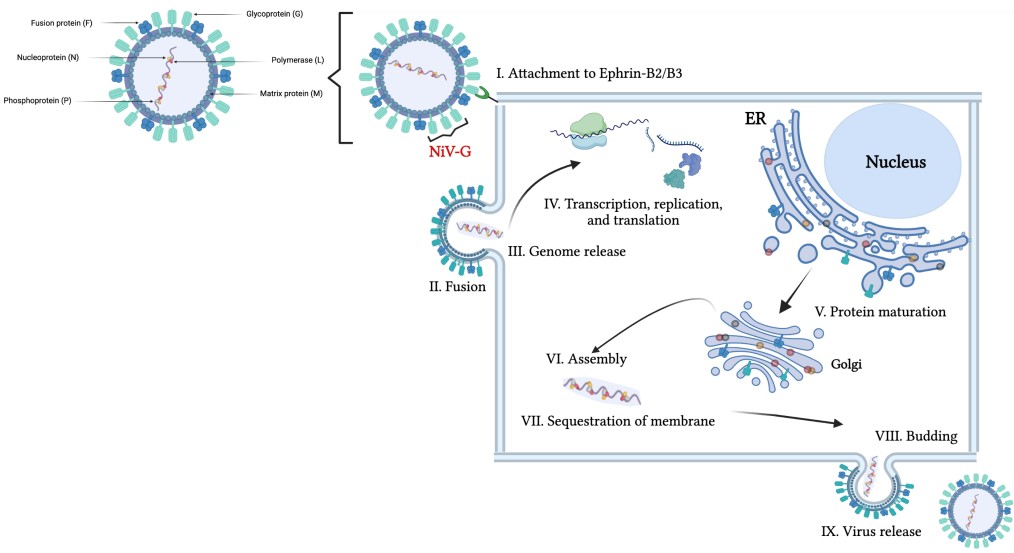

**Figure 1** **Schematic representation of the Nipah virus structure and its replication mechanism in host cells.** The left panel depicts the NiV's structural components. The right panel illustrates the viral replication process, which involves: (I) Attachment of NiV G to the host cell receptors, ephrin-B2/B3, initiating viral entry. The F protein facilitates merging of the viral and host cell membranes, allowing RNA release into cytoplasm (II and III). (IV) Transcription, replication, and translation involving P, L, and N proteins. The L polymerase protein, with P phosphoprotein as a cofactor, transcribes viral RNA into mRNA for translation. N protein ensures genome stability during replication. (V) Newly synthesized viral proteins undergo post-translational modifications, particularly glycosylation within the host's endoplasmic reticulum (ER) and Golgi for proper folding and function. Viral components are assembled, and the M matrix protein facilitates membrane sequestration, preparing the virions for release (VI and VII). New virions bud off from the host cell membrane, acquiring a lipid envelope, and are released to infect new host cells, propagating the infection cycle (VIII and IX) (*Hauser et al., 2021*; *Quarleri, Galvan & Delpino, 2022*). Figure created in Biorender (https://app.biorender.com). Adapted from *Meier et al. (2024)*; *Yang & Kar (2024)*.

*& Dutch, 2012*; *Navaratnarajah et al., 2020*; *Ortega et al., 2022*). The NiV RNA genome is encapsidated by the N nucleoprotein, forming a helical nucleocapsid assembly, which not only protects the RNA from nucleases degradation but also serves as a template for productive mRNA transcription and replication of the newly synthesized RNA genome by the RNA-dependent RNA polymerase L (*Omi-furutani et al., 2010*; *Ogino & Green, 2019*; *Peng et al., 2024*). The phosphoprotein acts as a cofactor that helps anchoring the RNA-dependent RNA polymerase to the nucleocapsid (*Chen, Ogino & Banerjee, 2006*; *Yabukarski et al., 2014*). The matrix protein is crucial for coordinating virion budding by organizing viral structural components at specific assembly sites on the host cell's plasma membrane (*Ciancanelli & Basler, 2006*; *Harrison, Sakaguchi & Schmitt, 2010*; *Patch et al., 2007*; *Watkinson & Lee, 2016*).

## NiV G glycoprotein structure

NiV and Hendra virus (HeV) are highly pathogenic members of the genus *Henipavirus*. Their G glycoproteins share ≥80% amino acid sequence identity, while their corresponding

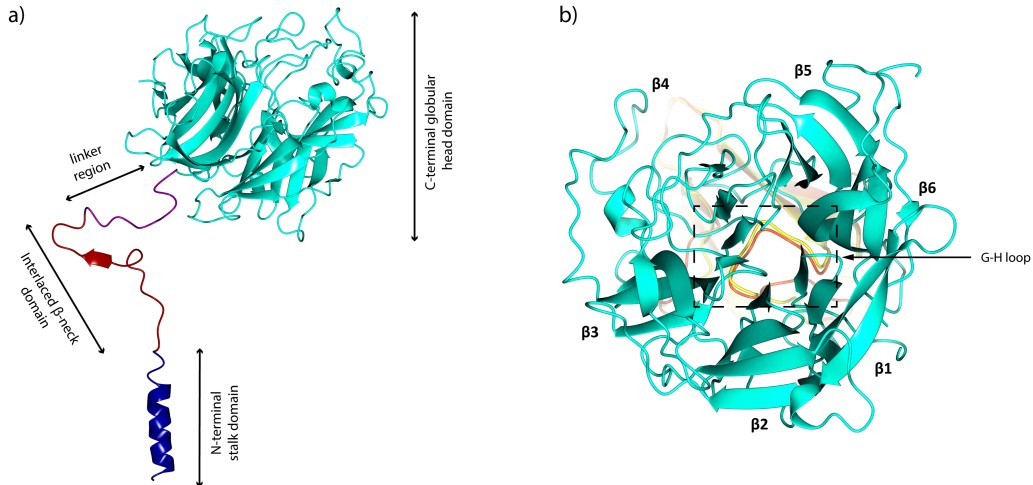

**Figure 2** **Structural organization of the NiV G glycoprotein ectodomain monomer.** (A) A single chain of the G glycoprotein which consists of N-terminal stalk domain (dark blue), the interlaced $\beta$-neck domain (dark red), the linker region (dark magenta), and the C-terminal six-bladed $\beta$-propeller globular head domain (cyan) (PDB: 7TXZ). (B) Top view of the NiV G glycoprotein head domain (cyan), which adopts a disk-like shape and is composed of six $\beta$-propeller blades ($\beta1$–$\beta6$). The ephrin-B2/B3 G-H loop is superimposed around the NiV G central cavity. The structural visualization was generated using CCP4mg (*McNicholas et al., 2011*).

nucleotide sequences exhibit 70.8% similarity (*Harcourt et al., 2000*). Structurally, both G glycoproteins are highly conserved, with a root-mean-square deviation (RMSD) of 0.5 Å in their C$\alpha$ atoms over 189–601 residues (*Bowden et al., 2008*). This substantial similarity results in significant antigenic cross-reactivity, providing valuable insight for diagnostic methods and vaccine development targeting both viruses (*Hsu, 2006*). Similar to the G glycoprotein from HeV, the NiV G glycoprotein is a type II integral membrane protein that consists of a homotetrameric ectodomain. Each monomer consists of several domains, including an N-terminal stalk domain (residues 96–147), an interlaced $\beta$-neck domain (residues 148–165), and a linker region (residues 166–177) that connects to the C-terminal six-bladed $\beta$-propeller globular head domain (residues 178–602) (Fig. 2). Additionally, the NiV G glycoprotein features a cytoplasmic tail that anchors the protein to the inner side of the viral envelope and a single transmembrane helix that spans the lipid bilayer, connecting the cytoplasmic tail to the ectodomain (*Bowden et al., 2008*; *Wang et al., 2022a*). The globular head domain contains specific binding sites that recognize and interact with ephrin-B2 and ephrin-B3, which are critical host cell receptors expressed on endothelial cells and neurons (*Bonaparte et al., 2005*; *Negrete et al., 2005*; *Negrete et al., 2006*; *Xu, Broder & Nikolov, 2012*). Although NiV and HeV share similar cellular tropisms and utilize the same receptor set, NiV binds to ephrin-B3 with 30-fold higher affinity than HeV, while both viruses exhibit similar binding affinities for ephrin-B2 (*Negrete et al., 2007*).

## NiV G glycoprotein structure in complex with ephrin-B2 and ephrin-B3

The structure of ephrin-B2 has been determined at a resolution of 1.92 Å (PDB: 1IKO) (*Toth et al., 2001*). It features a globular domain composed of an eight-stranded $\beta$-barrel, arranged in two sheets around a hydrophobic core. The $\beta$-barrel consists of a mix of parallel and antiparallel $\beta$-strands, adopting a Greek key topology. Additionally, two $\alpha$-helices and a $3_{10}$helix are interspersed among the $\beta$-strands. Ephrin-B2 contains two buried disulfide bonds: one between C65 and C104 that stabilizes $\beta$-strands, and another between C92 and C156, anchoring two helices at the top of the barrel to enhance and maintain structural stability. Moreover, ephrin-B3 is structurally similar to ephrin-B1 and ephrin-B2, with a RMSD of 1.5 Å (*Nikolov et al., 2005*) and contains approximately 40% amino acid sequence identity. However, ephrin-B2 and ephrin-B3 differ structurally from ephrin-B1 in the receptor binding G-H loop conformation. In ephrin-B2 and ephrin-B3, the G-H loop adopts a more rigid conformation upon binding to the NiV G glycoprotein (Fig. 3), facilitating stable interactions. In contrast, the G-H loop in ephrin-B1 is more flexible, resulting in incompatibility with the NiV G glycoprotein binding (*Toth et al., 2001*). The critical residue F120 of ephrin-B2 interacts with Y581, I588, Q559, and E579 within the hydrophobic region of the NiV G glycoprotein's central cavity through Van der Waals forces, ensuring a strong and specific interaction (*Bowden et al., 2008*). Moreover, L124 and W125 of ephrin-B2 are crucial for NiV binding, interacting with W504 and F458 on the hydrophobic surface of the NiV G glycoprotein (Fig. 4A) (*Negrete et al., 2006*). These interactions trigger structural changes in the hydrophobic region of the G-H loop in the ephrin-B2 and ephrin-B3 receptors, allowing it to bind within the central cavity of the NiV G glycoprotein $\beta$-propeller head domain (*Negrete et al., 2006*). Additionally, the binding pockets for key residues (Y120, P122, L124, and W125) of ephrin-B3 in the G-H loop region are formed through specific hydrophobic and polar interactions involving residues I588, I580, Y581, P488, V507, A532, T531, G489, Q490, E505, G506, W504, I401, F485, and L305, along with the C216–C240 disulfide bridge (*Xu et al., 2008*). These interactions stabilize the receptor-binding interface and trigger conformational changes in the NiV G glycoprotein (Fig. 4B) that bring the viral and host cell membranes into close proximity, thereby facilitating viral attachment and significantly enhancing the membrane fusion activity of the associated F glycoprotein (*Xu et al., 2008*). This fusion event allows the viral genome to be released into the host cell cytoplasm, where it hijacks the host's cellular machinery for replication (*Aguilar et al., 2010*; *Liu et al., 2014*). The interaction of the NiV G glycoprotein with host cell receptors involves protein-protein interactions, unlike other *Paramyxoviridae* members, such as Parainfluenza and Newcastle disease viruses, which rely on sialic acid attachment (*Bowden et al., 2008*). The surface area buried in the NiV G/ephrin-B2 complex is 2,800 Å$^2$ (*Bowden et al., 2008*), while in the NiV G/ephrin-B3 complex, it is 2,600 Å$^2$ (*Xu et al., 2008*). This results in a relatively flat binding interface compared to Parainfluenza and Newcastle disease viruses, where sialic acid binds much more deeply into the centre of the $\beta$-propeller globular head domain. Despite this, the NiV G glycoprotein retains a distinct cleft that is homologous to the sialic acid binding pocket. Furthermore, both ephrin-B2 and ephrin-B3 interact with Eph receptors, which belong to

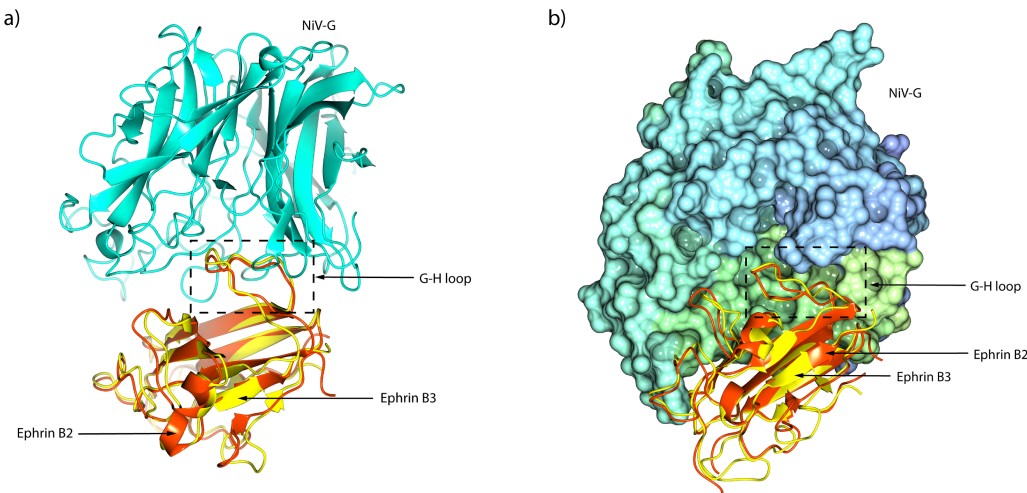

**Figure 3 The NiV G/ephrin-B2/B3 complexes.** (A) Side view of the overall structure of NiV G (cyan) bound to the superimposed G-H loops of ephrin-B2 (orange red) and ephrin-B3 (yellow). The G-H loop is positioned near the receptor-binding site of NiV G, playing a critical role in receptor recognition. (B) Surface representation of NiV G with the superimposed G-H loops of ephrin-B2 (orange red) and ephrin-B3 (yellow), illustrating the hydrophobic pocket involved in receptor engagement. The hydrophobic pocket of NiV G facilitates high-affinity binding by accommodating key residues of ephrinB2/B3, which is crucial for viral entry into host cells. The structural visualization was generated using CCP4mg (*McNicholas et al., 2011*).

the large receptor tyrosine kinase family (*Bowden et al., 2008*). These interactions facilitate bidirectional signalling, a key feature of ephrin-Eph receptor communication on the cell membrane surface (*Bradel-tretheway et al., 2019*; *Xu, Broder & Nikolov, 2012*).

## Structural and functional evaluation of the NiV G glycoprotein *via* mutagenesis studies

Mutagenesis studies on the NiV G glycoprotein are essential for anticipating genetic evolution, guiding the development of vaccines and therapies, and improving surveillance strategies, all of which contribute to preparedness for a potential NiV pandemic. Mutation in E533 and E505 significantly reduce the binding affinity between the NiV G glycoprotein and ephrin-B2. The E533Q mutation enables NiV to escape neutralization by the monoclonal antibody mAb 3B10 (*Guillaume et al., 2006a*), while also impairing the fusion-promoting activity of the NiV G glycoprotein. This results in more than a 50% reduction in viral fusion compared to the wild-type (*Guillaume et al., 2006b*). Moreover, the substitutions W504A, E505A, Q530A, T531A, A532K, and N557A each reduced the fusion-promoting activity of the NiV G glycoprotein by ≥50%. A comparative study on amino acid 507 of the HeV G and NiV G glycoproteins provides further insights into receptor usage. S507 in HeV significantly reduced ephrin-B3-dependent entry, nearly 10-fold lower than V507 in NiV. However, substituting serine with threonine at residue 507 in HeV restored ephrin-B3 receptor binding efficiency to a level comparable to V507 in NiV. These findings suggests that the shared hydrophobic methyl group in T507 (HeV) and V507 (NiV) is more critical for ephrin-B3 utilization than the polar similarity between

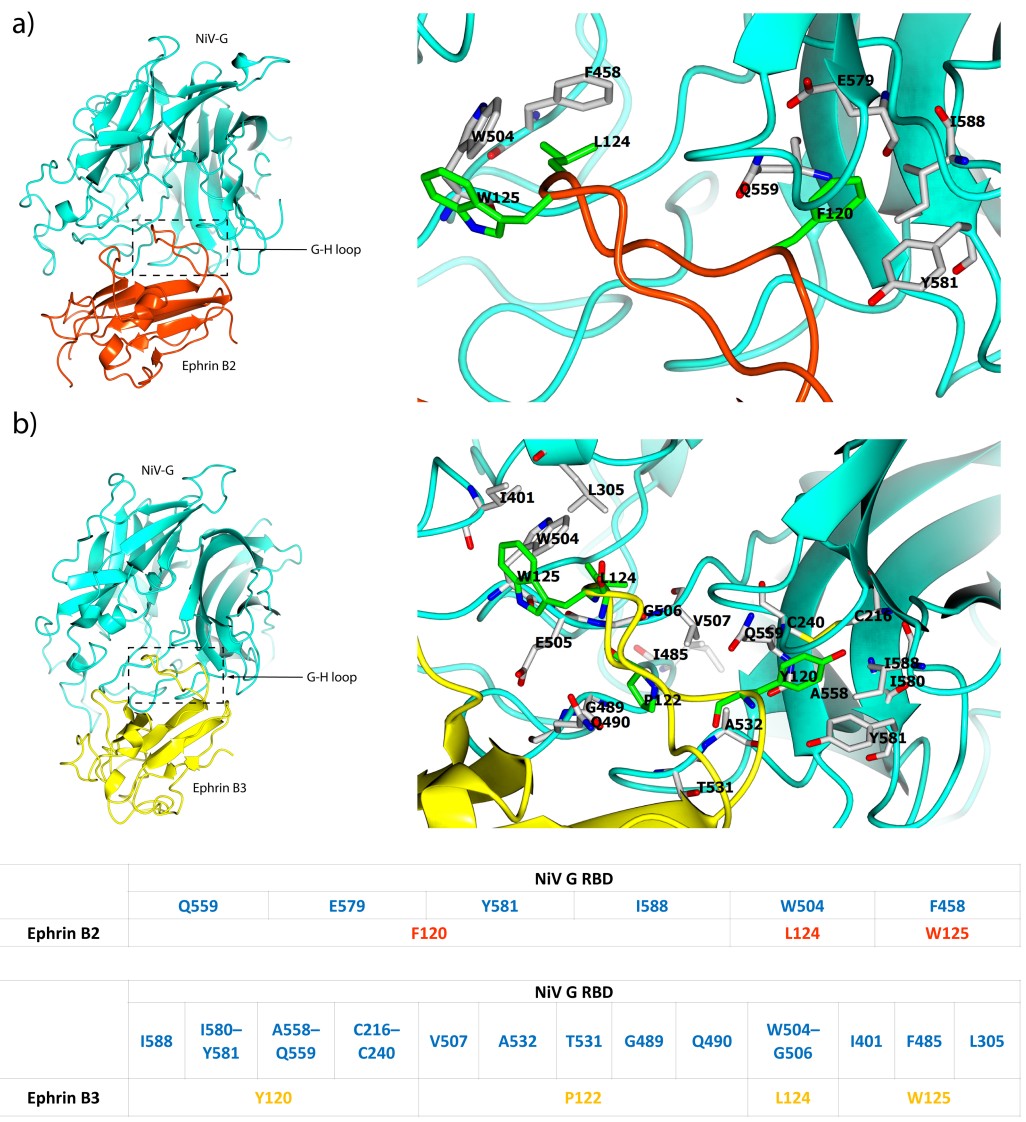

a)

b)

| | NiV G RBD | | | | | |
|---|---|---|---|---|---|---|
| | Q559 | E579 | Y581 | I588 | W504 | F458 |
| Ephrin B2 | F120 | | | | L124 | W125 |

| | NiV G RBD | | | | | | | | | | | | |
|---|---|---|---|---|---|---|---|---|---|---|---|---|---|
| | I588 | I580–Y581 | A558–Q559 | C216–C240 | V507 | A532 | T531 | G489 | Q490 | W504–G506 | I401 | F485 | L305 |
| Ephrin B3 | Y120 | | | | P122 | | | | | L124 | | W125 | |

**Figure 4** **The NiV G/ephrin-B2/B3 complexes.** (A) Left: overall structure of NiV G (cyan) bound to ephrin-B2 (orange red) (PDB:2VSM). Right: interacting residues are labelled. The position of the G-H ephrin-B2 loop is shown to illustrate the binding pockets residues F120, L124, and W125. (B) Left: overall structure of NiV G (cyan) bound to ephrin-B3 (yellow) (PDB:3D12). Right: interacting residues are labelled. The position of the G-H ephrin-B3 loop is shown to illustrate the binding pockets residues Y120, P122, L124, and W125. The Y120 binding pocket is only formed upon ephrin binding (*Xu et al., 2008*). Bottom panel: the table summarises key amino acid residues in the G-H loop of ephrin-B2 and ephrin-B3 that interact with the NiV G receptor-binding domain (RBD) through hydrophobic and polar interactions, as well as a disulfide bridge. The structural visualization was generated using CCP4mg (*McNicholas et al., 2011*).

serine and threonine (*Negrete et al., 2007*). Furthermore, the E533Q mutation has a similar effect, completely abolishing ephrin-B2 binding. Additionally, the E533Q, E505A, W504A, and V507S mutations significantly reduced ephrin-B3 binding (*Guillaume et al., 2006b*; *Negrete et al., 2006*). Moving from the receptor-binding head domain to the stalk region,

structural integrity also proves vital for viral function. The stalks cysteine residues of the Henipavirus G glycoprotein are located adjacent to a proline-rich microdomain, a feature unique to the *Henipavirus* genus. These cysteine residues play a crucial role in maintaining oligomeric stability and are essential for triggering fusion (*Maar et al., 2012*). A series of residues in the NiV G glycoprotein stalk region were substituted with cysteines (Table 1) to enhance tetrameric structural stability through disulphide bond interactions. However, this mutation significantly reduces cell–cell fusion without affecting cell surface expression or ephrin-B2 binding affinity (*Ortega et al., 2022*). A molecular modelling study using single-point site-directed mutagenesis suggests that increased tetrameric stability limits the mobility of the G tetrameric stalk structure, significantly impairing its ability to trigger the fusion F glycoprotein and thereby directly reducing membrane fusion. Expanding on cross-species insights, studies comparing NiV and Ghana virus (GhV) provide further evidence of how specific G glycoprotein domains govern receptor usage. While NiV uses both ephrin-B2 and B3 as entry receptor, GhV exclusively utilizes ephrin-B3 (*Oguntuyo et al., 2024*). A chimeric structural model, in which the head domains of NiV and GhV were exchanged, was constructed to investigate the regions responsible for their differential usage of ephrin-B2 and ephrin-B3. Interestingly, the study found that a chimera constructed with the stalk domain of NiV and the head domain of GhV completely abolished ephrin-B3 binding. Further structure-informed mutagenesis analysis based on this chimera identified the N557S and Y581T mutations in the head domain of the NiV G glycoprotein as significantly impairing its ability to bind ephrin-B3 (*Oguntuyo et al., 2024*). Additionally, it has been identified that Y120 is critical for the ephrin-B3 receptor usage (*Oguntuyo et al., 2024*). The N557S and Y581T mutations in NiV disrupt its ability to $\pi$-stack with Y120 of ephrin-B3, as these amino acids have polar, neutral side chains. This alteration directly abolishes binding and prevents the usage of ephrin-B3. Across these mutagenesis studies, which reveal critical determinants of receptor specificity and fusion activity, a comprehensive understanding of the structural biology of the G and F glycoproteins provides valuable insights into the rational design of vaccines and therapeutics (*May & Acharya, 2024*; *Salleh, 2025*).

## Structural and functional evaluation of the NiV G glycoprotein *via* conformational antibodies

Crucially, the development of vaccines and therapeutics for Henipavirus infections depends on understanding its molecular mechanisms and cell entry processes, which are mediated by the G and F glycoproteins. The ectodomain G glycoprotein, also referred to as the receptor binding protein (RBP) is responsible for binding to host cell receptors ephrin-B2/B3. In addition to structure-based mutagenesis studies, conformational changes in the NiV G glycoprotein can also be detected using conformational antibodies such as mAb213, mAb45, and mAb167. These antibodies target distinct structural regions different from those recognized by ephrin-B2-competing antibodies like m102.4 and HENV-26 (*Xu et al., 2013*; *Dong et al., 2020*; *Wang et al., 2024*). These antibodies serve as valuable tools for studying the structural dynamics of the NiV G glycoprotein and identifying mutations that may impact vaccine and therapeutic efficacy (Table 2). Notably, potent

Mohamad Nasir et al. (2025), *PeerJ*, DOI 10.7717/peerj.19835

**Table 1** Summary of key mutations in the NiV G glycoprotein and their functional implications.

| No | Key mutations | NiV G Domain | Animal model study/ Cells | Receptor proteins | Structural and functional effects on the NiV G | Mutagenesis strategies | Reference |
|---|---|---|---|---|---|---|---|
| 1. | E533Q E504A W505A Q530A T531A A532K N557A | Globular head | African green monkey/Vero E6 cells, Hamster/ Chinese hamster ovary (CHO) cells | ephrin-B2 | • Reduce fusion promotion with surface expression comparable to wild type NiV G. | Mutate charged residues in the conserved region of NiV G and HeV G globular heads and generate NiV variants that escaped mAbs neutralization through sequencing. | *Guillaume et al. (2006a)*; *Guillaume et al. (2006b)* |
| 2. | W504A E505A V507S E533Q | Globular head | Hamster/ CHOpgsA745 mutant cells | ephrin-B2 & ephrin-B3 | • Reduction in ephrin-B3 binding. | NiV mutants were pseudotype onto vesicular stomatitis virus (VSV) reporter viruses and infected to CHO-B2 and CHO-B3 cells. | *Negrete et al. (2007)* |
| 3. | C146S C158S C162S | Stalk | Human/ HEK293T cells | ephrin-B2 | • Reduce oligomeric stability of NiV G and F fusion protein triggering. • Exhibited constitutive exposure of the mAb receptor binding-enhanced epitope. | NiV mutants were pseudotype onto VSV reporter viruses and infected to 293T cells. | *Maar et al. (2012)* |
| 4. | S179C S171C S137C S132C G125C S115C S110C T103C A86C S76C | Stalk | Human/ HEK293T cells | ephrin-B2 | • Increase tetrameric G structural stability and strength. • Reduce fusion promotion. | Disulphide bonds between G monomer stalk domains was created *via*-site directed mutagenesis to increase oligomeric strength and restricting the mobility of the specific region in the G stalk domains. | *Ortega et al. (2022)* |

Mohamad Nasir et al. (2025), *PeerJ*, DOI 10.7717/peerj.19835

**Table 1** (*continued*)

| No | Key mutations | NiV G Domain | Animal model study/ Cells | Receptor proteins | Structural and functional effects on the NiV G | Mutagenesis strategies | Reference |
|---|---|---|---|---|---|---|---|
| 5. | N557S Y581T | Globular head | Mouse/ HEK293T and U87 cells | ephrin-B3 | • Reduction in ephrin-B3 mediated entry. | Systematic, structure-informed mutagenesis to identify receptor-interfacing residues between NiV and GhV. | *Oguntuyo et al. (2024)* |

human monoclonal antibodies (mAbs) such as HENV-26 and HENV-32 have been shown to recognize diverse sites on the RBP of both NiV and HeV, demonstrating therapeutic potential in ferret models (*Dong et al., 2020*). The solved crystal structure of HENV-26/HeV-RBP (PDB: 6VY6) and HENV-26/NiV-RBP (PDB: 6VY5) have identified key RBP residues involved in antibody interactions. Specifically, residues V502 and D555–Q559 in HeV, as well as I502, P403, D555–Q559 in NiV, play crucial roles in binding. Additionally, the HENV-32/HeV-RBP complex (PDB: 6VY4) has revealed that residues P200, L202, F593, I203, Y205, V262, and P263 in HeV RBP contribute to hydrophobic interactions with HENV-32, underscoring their importance in antibody recognition and as potential targets for antibody optimization. However, the N557A mutation has shown to reduce viral fusion, while the N557S mutation affects binding affinity to ephrin-B3 (*Guillaume et al., 2006b*; *Oguntuyo et al., 2024*). Among the available therapeutic options, m102.4 remains the most promising post-exposure therapeutic, showing high effectiveness in animal models and strong potential for clinical use. m102.4 effectively neutralizes NiV and HeV by competitively inhibiting G-mediated viral attachment to the host receptors ephrin-B2 or ephrin-B3 (*Dang et al., 2021*). The m102.4 antibody targets highly conserved epitopes across viral variants, reducing the likelihood of viral escape due to mutations, even under therapeutic dosing in an immunocompetent host (*Playford et al., 2020*). However, an *in vitro*-engineered mutant of the NiV and HeV G glycoproteins that evaded m102.4 neutralization was found to carry a single amino acid substitution at positions V507I in NiV and D582N in HeV (*Xu et al., 2013*). Additionally, the nAH1.3 escape mutant in the NiV Malaysia strain G glycoprotein contained a Q450K mutation. Both the mAb hAH5.1 and 213 neutralization-escape mutants carried the N159D mutation. Furthermore, the hAH5.1 escape mutant had an R516K mutation, while the mAb 213 escape mutant carried a Q388R mutation (*Xu et al., 2013*).

## Functional implications of the NiV G glycoprotein mutations on the viral entry and host specificity

NiV exhibits broad host tropism, infecting a wide range of mammalian species, from fruit bats to pigs and humans (*Sahay et al., 2020*). This ability is attributed to the virus's reliance on the ephrin-B2 and ephrin-B3 receptors, which are highly conserved across mammalian species. These receptors are highly expressed in specific tissues, such as neurons and endothelial cells, which explains NiV's tissue tropism and associated pathologies. Unlike viruses that utilize sialic acid-mediated attachment, enabling infection across hosts with diverse glycan profiles, NiV specifically targets cells with abundant ephrin-B2 or ephrin-B3 expression (*Hooper et al., 2001*; *Wong et al., 2002*; *Palmer & Klein, 2003*; *Poliakov, Cotrina & Wilkinson, 2004*). Consequently, this specificity enables the virus to efficiently invade the central nervous system, leading to neurological disorders and vascular damage. The residues responsible for interacting with the viral protein are highly conserved between the ephrin-B2 and ephrin-B3 receptors across mammalian species, allowing NiV to bypass host-specific barriers and facilitating zoonotic transmission and multi-species infectivity (*Bossart et al., 2008*). The engineered mutations in the NiV G glycoprotein have provided crucial insights into its role in viral entry and host specificity. Mutations in the globular

**Table 2  Summary of key mutations in the NiV G glycoprotein targeted by neutralizing antibodies.**

| No | Neutralizing antibody | Epitope target region | Key mutations involved | Structural and functional effects of mutations on the NiV G/HeV G | Functional effects of the mutations on the neutralizing antibody | Reference |
|---|---|---|---|---|---|---|
| 1. | mAb213 | Distinct from ephrin-B2 binding sites | Q388R, N159D | Alters pre-receptor-binding conformation of G protein | Escape from neutralization (NiV and HeV) | *Xu et al. (2013)*; *Liu et al. (2014)*; *Borisevich et al. (2015)* |
| 2. | HENV-26 | Competes directly with ephrin B2/B3 binding sites | N557A (NiV), N557S (HeV) | Disrupts fusion triggering and reduces ephrin-B3 binding | Escape from neutralization (NiV and HeV) | *Guillaume et al. (2006b)*; *Xu et al. (2013)*; *Borisevich et al. (2015)*; *Dong et al. (2020)*; *Oguntuyo et al. (2024)* |
| 3. | m102.3/ m102.4 | Competes directly with ephrin-B2/B3 binding sites | V507I (NiV), D582N (HeV) | Increase affinity of the G proteins to both antibodies and ephrin-B2 by mutation V507I in NiV. Decrease affinity of the G proteins to both antibodies and ephrin-B2 by mutation D582N in HeV | Escape from neutralization (NiV and HeV) | *Xu et al. (2013)*; *Borisevich et al. (2015)* |
| 4. | hAH5.1 | Overlaps with or adjacent to ephrin-B2/B3 binding sites | R516K, N159D | Minor structural shifts; receptor binding remains intact | Escape from neutralization (NiV Malaysia strain) | *Xu et al. (2013)*; *Borisevich et al. (2015)* |
| 5. | nAH1.3 | Overlaps with or adjacent to ephrin-B2/B3 binding sites | Q450K | Disrupts local conformation within receptor-binding domain | Escape from neutralization (NiV Malaysia strain) | *Xu et al. (2013)*; *Borisevich et al. (2015)* |

head region, such as E533Q, E504A, W505A, Q530A, T531A, A532K, and N557A, have been shown to reduce fusion promotion with ephrin-B2 (*Guillaume et al., 2006b*). Additionally, mutations like W504A, E505A, V507S, E533Q, N557S, and Y581T in the globular head specifically impair ephrin-B3 binding, indicating their role in receptor selectivity and underscoring the structural determinants of host specificity (*Negrete et al., 2007*; *Oguntuyo et al., 2024*). Structural alterations in the stalk domain also significantly impact viral entry by destabilizing NiV G oligomerization, leading to constitutive exposure of receptor-binding epitopes and altered F fusion activity (*Maar et al., 2012*). Conversely, engineered disulfide bonds between stalk monomers increase tetrameric G stability while restricting mobility, thereby reducing fusion promotion (*Ortega et al., 2022*). These engineered modifications demonstrate that the NiV G glycoprotein is highly adaptable, with specific residues governing receptor interaction, fusion efficiency, and species tropism. Understanding these functional changes provides valuable insights into the mechanisms of cross-species transmission and contributes to the development of therapeutic strategies targeting receptor binding.

## Functional implications of the NiV G glycoprotein mutations on the vaccine and therapeutics

The mutational insights highlight how structural alterations in the NiV G glycoprotein can significantly affect the design of vaccines and therapeutics. Specific mutations such as E533Q, E505A, W504A, and V507S drastically reduce ephrin-B2 and B3 binding affinity and impair fusion activity by more than 50%, while also enabling immune escape from monoclonal antibody mAb 3B10 (*Guillaume et al., 2006a*; *Guillaume et al., 2006b*; *Negrete et al., 2006*). The D582N mutation, located in the $\beta$6S2–S3 region outside of the receptor or mAb binding interface, affects the interaction between Fab and G protein. In the wild-type HeV structure, D582 forms salt-bridges with R589 and K591 on the $\beta$6S3 strand. The substitution of aspartic acid with asparagine at this point likely induces a conformational change in the $\beta$6S3 region, leading to a rearrangement of key m102.3-contacting residues, including I580, Y581, and I588. This structural shift may hinder the interaction of both the antibody and the NiV G-H loops (*Xu et al., 2013*). Moreover, mutations like N557S and Y581T disrupt $\pi$-stacking interactions, crucial for ephrin-B3 recognition, directly abolishing receptor usage (*Oguntuyo et al., 2024*). These findings suggest that even minor alterations in the RBD can severely affect functional outcomes. Notably, although mutations in the stalk domain do not affect cell surface expression or ligand accessibility (*Ortega et al., 2022*), they significantly alter the conformational dynamics of the G glycoprotein, underscoring the complexity of epitope targeting. Understanding these structural changes is crucial for predicting the likelihood of NiV strains developing resistance to immune responses, whether naturally acquired or vaccine induced. This highlights a key challenge for antibody-based therapies, which mutations can alter or eliminate epitopes, reducing the efficacy of neutralizing antibodies (*Borisevich et al., 2015*; *Wang et al., 2022a*; *Larsen et al, 2025*). This knowledge significantly contributes to the design of new antibody therapies that target conserved epitopes. As a result, there is an urgent need to develop broadly neutralizing antibodies that can target conserved regions unaffected by mutations. In line with this, a recent study has identified a unique NiV-neutralizing single-domain antibody (UdAb), n425, that specifically targets a conserved cryptic epitope located at the dimeric interface of the NiV G glycoprotein. This antibody demonstrates cross-neutralizing potential, shows promise as a candidate for informing the development of a universal NiV-HeV vaccine (*Wang et al., 2024*). Compared to the full-length mAb m102.4, n425 shows significantly higher potency in inhibiting viral membrane fusion and demonstrates more efficient penetration into the murine brain, suggesting improved therapeutic potential against NiV-associated neurological complications. Interestingly, n425 targets a cryptic epitope at the dimeric interface of the NiV G glycoprotein. Although HENV-32 shares an overlapping epitope with n425, the two antibodies exhibit significantly different binding engagements. Structural analyses of the HENV-32/H eV-RBP complex (PDB ID: 6VY4) showing that the HENV-32 interacts primarily with the N-terminal T196–I209 segment and the $\beta$1-strand region of the epitope (*Dong et al., 2020*), whereas n245 additionally binds to the $\beta$2-strand region (*Wang et al., 2024*). Moreover, only two amino acid residues differ among HeV and the NiV Malaysia and Bangladesh strains within the binding epitope of n245. This is fewer than the residue differences observed in the epitopes targeted by m102.4, HENV-26,

hAH1.3, nAH1.3, and HENV-32 (*Zhu et al., 2006*; *Bossart et al., 2009*; *Geisbert et al., 2014*; *Charlier et al., 2022*; *Wang et al., 2022b*), highlighting n245 as a promising candidate for therapeutic application and a potential guide for universal vaccine design (*Wang et al., 2024*). This mechanism is reminiscent of hidden epitopes found at the trimeric interface of hemagglutinin in the influenza virus (*Bangaru et al., 2020*) and the spike protein in SARS-CoV-2 (*Li et al., 2022*), where antibodies binding to the conserved trimeric interface of the SARS-CoV-2 spike protein have been shown to trigger its disassembly, thereby enhancing viral neutralization. Similarly, n425 binds to the dimeric interface of the NiV G glycoprotein, disrupting its tetramerization and effectively preventing F protein activation, which is crucial for viral entry (*Wang et al., 2024*). These insights align with structural stabilization findings, where cysteine substitutions in the NiV G stalk domain enhance tetrameric stability but reduce fusion integrity (*Ortega et al., 2022*), suggesting that such conformational constraints could also be exploited therapeutically. Moreover, N-linked glycosylation sites, genetically encoded through the conserved $N - X - S/T$ sequon can confer selective advantages by sterically masking key antigenic epitopes, thereby promoting immune evasion (*Miller et al., 2021*; *Alves et al., 2022*). Notably, the Malaysia strain of NiV G glycoprotein contains 28 N-glycan sites per tetramer, while its F fusion protein exhibits 15 per trimer (*Hawkins et al., 2025*). These glycan-mediated adaptations are often preserved through positive selection, particularly when they do not compromise receptor binding or overall viral fitness (*Hawkins et al., 2025*). While most N-glycan sites are conserved across NiV strains, an identified N-glycosylation site N481 on the G glycoprotein showing evolutionary distinct, presenting a specific mutation N481D that results in the loss of an N-glycan site (*Hawkins et al., 2025*). While the N481D mutation itself has a minimal impact on the conformational dynamics and receptor binding, the study emphasizes the important of a conserve loop region (T483–F496) near the N481 site for ephrin-B2 binding as it is conserved across all NiV strains and the primary strain for HeV, which significantly make it a relevant consideration for vaccine and therapeutic design. The mutation has shown a minimal impact on conformational dynamics of the conserve loop region and on the receptor binding, suggesting it may not directly affect infectivity (*Hawkins et al., 2025*). Altogether, mutagenesis studies not only deepen our understanding of structure-function relationships but also provide a vital foundation for guiding vaccine design and the development of next-generation therapeutics.

## CONCLUSIONS

Mutational analyses, particularly those involving engineered mutations of the NiV G glycoprotein reveal its critical role in mediating viral entry, host specificity, and immune evasion. Specific amino acid residues within the globular head and stalk domains modulate receptor binding affinity, fusion efficiency, and oligomeric stability, ultimately influencing zoonotic transmission and pathogenesis. Although NiV is an RNA virus, it has accumulated relatively few naturally occurring mutations compared to viruses like SARS-CoV-2. Notably, a recently documented natural mutation, N481D, located at an N-glycosylation site on the G glycoprotein, shows minimal impact on conformational dynamics of the

conserve loop region and receptor binding, suggesting it may not directly affect infectivity. This genomic stability and strong selective pressure underscore the importance of continued structural and functional surveillance to inform vaccine and therapeutic development. Importantly, mutagenesis studies have shown that certain mutations can impair ephrin-B2/B3 engagement or disrupt antibody binding, highlighting the challenge of immune escape. However, the functional consequences of these mutations in the context of natural infection remain poorly understood. Future research should prioritize *in vivo* validation of these mutational effects across diverse host models, along with structural studies of naturally emerging NiV variants. Additionally, exploring conserved, glycan-shield epitopes such as the cryptic, conserved dimeric interface targeted by the potent single-domain antibody n425, offers promising avenues for broad-spectrum vaccine and therapeutic development. If NiV were to acquire efficient human-to-human transmission similar to SARS-CoV-2, it could lead to a devastating pandemic, given the current absence of approved vaccines or therapeutic agents. This potential scenario is particularly concerning, making it a much more dangerous threat if it were to evolve increased transmissibility. However, as a high-risk pathogen requiring BSL-4 containment, research on NiV variants remain significantly limited and underreported. These restrictions hinder large-scale virological studies, particularly those involving live virus infections and long-term surveillance of viral evolution. Despite these challenges, mutagenesis studies on the NiV G glycoprotein offer valuable insights into its functional consequences and directly contribute to the development of effective vaccines and therapeutics.

## ACKNOWLEDGEMENTS

I acknowledge the use of chatGPT to refine the academic language and check for grammatical error of my own work.

### Funding

This work was supported by the Kementerian Pendidikan Tinggi Malaysia, Fundamental Research Grant Scheme (Project No: FRGS/1/2024/SKK12/USM/03/1) and Universiti Sains Malaysia, Short-Term Grant (Project No: R501-LR-RND002-0000000996-0000). The funders had no role in study design, data collection and analysis, decision to publish, or preparation of the manuscript.

### Grant Disclosures

The following grant information was disclosed by the authors:
Kementerian Pendidikan Tinggi Malaysia: FRGS/1/2024/SKK12/USM/03/1.
Universiti Sains Malaysia: R501-LR-RND002-0000000996-0000.

### Competing Interests

The authors declare there are no competing interests.

## Author Contributions

- Nur Syafiqah Mohamad Nasir conceived and designed the experiments, performed the experiments, analyzed the data, prepared figures and/or tables, authored or reviewed drafts of the article, and approved the final draft.
- Yasmin Khairani Muhammad Ismadi conceived and designed the experiments, performed the experiments, analyzed the data, authored or reviewed drafts of the article, and approved the final draft.
- Noreafifah Semail conceived and designed the experiments, performed the experiments, analyzed the data, prepared figures and/or tables, authored or reviewed drafts of the article, and approved the final draft.
- Wan Alif Syazwani Wan Alias conceived and designed the experiments, performed the experiments, analyzed the data, authored or reviewed drafts of the article, and approved the final draft.
- Nik Mohd Noor Nik Zuraina analyzed the data, authored or reviewed drafts of the article, and approved the final draft.
- Nik Yusnoraini Yusof analyzed the data, authored or reviewed drafts of the article, and approved the final draft.
- Zakuan Zainy Deris analyzed the data, authored or reviewed drafts of the article, study supervision, and approved the final draft.
- Mohd Zulkifli Salleh analyzed the data, authored or reviewed drafts of the article, obtained funding, study supervision, and approved the final draft.

## Data Availability

This is a literature review.

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
