# Peer review of "Antigenic and mutational insights into the Nipah virus G glycoprotein: implications for viral entry, host specificity, therapeutics, and vaccine development"

_PeerJ, doi:10.7717/peerj.19835_

## Round 0.1 · original submission · Major Revisions

· Academic Editor

Major Revisions

The review by Nasir et al., provides a comprehensive overview of engineered mutations in the Nipah virus (NiV) G glycoprotein and their implications for viral entry, host specificity, vaccines, and therapeutics. This topic is highly relevant due to NiV's classification as a BSL-4 pathogen and the lack of approved vaccines or treatments. The manuscript is interesting and well-structured.

I have a few comments listed below:

1) The manuscript highlights the limited reports on naturally occurring NiV strains with significant mutations in their surface glycoproteins, especially when compared to the thousands of documented mutations in viruses like SARS-CoV-2. While the conclusion mentions this difference, the authors could explore more why the NiV G glycoprotein does not change as much. Since NiV is an RNA virus that usually has a high mutation rate, is it possible that strong pressure to keep binding to the important ephrin-B2/B3 receptors explains this? Or could it be that natural mutations are simply not reported enough because of BSL-4 lab rules? Expanding on this topic would help the readers to understand the importance of engineered mutations better.

2) It’s important to be clear about what the authors mean by "receptor binding protein (RBP)." In the antibody section, this term specifically means the ectodomain of the G glycoprotein, which is the part that connects to receptors. By stating early on that the G glycoprotein is the "receptor binding protein" (specifically its ectodomain), the authors can help avoid confusion for readers who may not know the specific terms related to henipaviruses.

3) The review offers valuable insights into engineered mutations in the receptor binding and stalk domains. However, it would be beneficial to include studies on mutations in other NiV G domains, such as the β-neck domain, linker region, cytoplasmic tail, and transmembrane helix. If no data exists for these areas, noting this gap would be helpful. Highlighting them as potential topics for future research could also add value.

4) The manuscript is mostly clear, but some transitions between paragraphs need improvement. This is particularly true in the sections "Structural and functional evaluation of the NiV G glycoprotein via mutagenesis studies" and "Functional implications of the engineered NiV G glycoprotein on the vaccine and therapeutics."

Reviewer 1 ·

Basic reporting

In general the authors create a reasonable review on Nipah Virus G protein antigenicity. However there are still some concerns need to be resolved.

Major:

The title may need to change as this review is mainly trying to summarize current antigenicity studies on NiV G protein rather than summarize current works on how to engineer G protein.

Line 321-322: Most escape mutations will not lead to antigen conformation change, please cite properly if there are clear evidence that NiV G has escape mutations that are not located at corresponding epitope and reduce mAbs binding by epitope shielding.

Line 334-342: HENV-32 binds to the dimeric interface.

Line 364-373: Current summarized studies are on G protein antigenicity, receptor tropism and escape mutations, which is not about “engineered NiV G”

Minior:

Line 173-176: “These interactions trigger structural changes......” needs citation

Line 181-183: “This interaction......” It seems currently there’s no direct evidence that receptor binding of G protein will have “conformational changes that bring the viral and host cell membranes into close proximity”. Please cite if there’s evidence, or please rewind the sentence to remove the implied active “conformational changes”

Experimental design

Not applicable as this is a review.

Validity of the findings

Not applicable as this is a review.

Reviewer 2 ·

Basic reporting

.

Experimental design

.

Validity of the findings

.

Additional comments

Title of the article: Structural insights into the engineered G glycoprotein of Nipah virus: Implications for viral entry, host specificity, vaccines, and therapeutics

Major comments:
1. The manuscript is well-written and provides a comprehensive overview of structural data related to the Nipah virus G glycoprotein
2. The discussion effectively connects structural insights with implications in monitoring viral evolution, host specificity, and in the development of vaccines and therapeutics.
3. The figures and tables are well-represented.
4. The ‘Abstract’ would benefit from being revised to focus more closely on the review’s core topic.
5. The search-methodology paragraph could be expanded to include the exact Boolean strings used and the dates searched.
6. The review “Structural Studies of Henipavirus Glycoproteins” by May & Acharya (2024, Viruses 16:195) is to be cited, along with a brief statement clarifying the distinct contribution of the present work.
7. In ‘Conclusions’, placing greater emphasis on the review’s key insights and less on pandemic risk would strengthen the overall coherence of the manuscript.

Minor comments:
1. Line 133: The statement "glycoproteins share ≥80% homology" should be clarified. It is important to specify whether the figure refers to sequence identity or sequence homology, as these terms are not interchangeable.
2. Line 343: The phrase "nAb425 differs by only two amino acid residues" should be revised to more accurately reflect the context. Suggested revision: "The epitope of nAb425 differs by only two amino acid residues."
3. Consider rewriting long sentences for clarity, e.g. lines 68-71, 216-219, 356-359.
4. The authors are suggested to change subjective terms like “extremely similar” (line 134) correct grammar (e.g., line 180: use “stabilize” instead of “stabilizing”), and fix article usage (line 362: “a biosafety level-4 laboratory”).
5. Consider adding a summary table that lists each engineered residue, its structural context, experimental system, and effect on receptor usage, fusion, or antibody escape.
6. The manuscript would benefit from the addition of a figure or table summarizing key residues in the G glycoprotein that are targeted by neutralizing antibodies.
7. While the authors have extensively covered engineered mutations in the NiV G glycoprotein, it would strengthen the review to include a discussion of mutations at N-glycosylation sites, which are known to influence viral fusion activity and sensitivity to neutralizing antibodies.
8. The recent preprint by Hawkins et al., 2025, which analyses natural variations in NiV-G N-glycosylation sites may be cited.

Reviewer 3 ·

Basic reporting

1. The manuscript uses professional and unambiguous English throughout. Minor stylistic edits could improve sentence flow, but overall language quality is acceptable.

2. The background information provided is comprehensive, current, and supported by appropriate literature references. The review properly establishes the role of the Nipah virus G glycoprotein in viral entry, pathogenesis, and vaccine development.

3. The manuscript is well formatted with well-arranged paragraphs and proper use of tables and figures. Raw data sharing is not, however, available for this type of article, but all the sources are well referenced and accessible.

4. The manuscript is of interest to virology in general, structural biology, and drug development. The topic has been recently reviewed, but the review is value-added since it is focused on engineered mutations and what this implies, which is enough to merit its contribution.

5. The introduction properly identifies the aim of the review and defines the audience.

Experimental design

1. The article is consistent with the scope and aims of PeerJ and is consistent with expectations for a literature review.

2. Search strategy is adequately explained, including databases and keywords. More specific reference to inclusion/exclusion criteria, and how studies were selected and screened, would improve the manuscript.

3. Sources are properly referenced, and references are paraphrased and smoothly inserted into the text.

4. The organization is clear, and there is a logical progression from structural biology to functional significance and therapeutic approaches. There are useful subheadings and clarity throughout.

Validity of the findings

1. Arguments made are well supported by literature cited. Authors connect findings in several studies well, tracing structural findings to functional outcomes of viral entry and immune escape.

2. The conclusions are consistent with the information presented in the body of the review and directly address the targets as presented in the introduction.

3. The review outlines the critical open questions, such as the lack of structural data, poor information on naturally occurring mutations, and the limitations imposed by BSL-4 stipulations. The way forward is well set.

Additional comments

1. The paper presents an equal mix of the latest developments in Nipah virus G glycoprotein research.

2. While the tables and figures showing extensive mutations and their functional consequences are provided, the resolution and quality of the figures in certain instances are not sufficient because many of them look pixelated. Good-quality clear images are required so that the reader can properly understand the intricate molecular information.

3. Language editing software credit is accurate and clear.


Minor corrections: The paper is well written and scientifically sound but would be enhanced by making the process of study selection clearer and slight amplification in figure clarity and sentence construction. Re-review is not warranted after these changes.

---

## Round 0.2 · accepted · Accept

· Academic Editor

Accept

The authors have satisfactorily addressed all reviewer and editor comments and substantially revised the manuscript. The revised version now fully meets the journal's standards and is suitable for publication

Reviewer 1 ·

Basic reporting

Thanks for the authors to address all comments.

Experimental design

Thanks for the authors to address all comments.

Validity of the findings

Thanks for the authors to address all comments.

Reviewer 2 ·

Basic reporting

No comment. The revised manuscript is clear and well organised.

Experimental design

No comment. The authors have addressed all previous comments

Validity of the findings

No comments. Conclusion has been rewritten and fully address the earlier comments

Additional comments

No comment. All earlier concerns have been fully resolved.